# Multi-Objective PSO with Variable Number of Dimensions for Space Robot Path Optimization

**Petr Kadlec** 

Lerch Laboratory of EM Research, Department of Radio Electronics, Faculty of Electrical Engineering and Communication, Brno University of Technology, Technická 3082/12, 616 00 Brno, Czech Republic; kadlecp@vut.cz

**Abstract:** This paper aims to solve the space robot pathfinding problem, formulated as a multi-objective (MO) optimization problem with a variable number of dimensions (VND). This formulation enables the search and comparison of potential solutions with different model complexities within a single optimization run. A novel VND MO algorithm based on the well-known particle swarm optimization (PSO) algorithm is introduced and thoroughly described in this paper. The novel VNDMOPSO algorithm is validated on a set of 21 benchmark problems with different dimensionality settings and compared with two other state-of-the-art VND MO algorithms. Then, it is applied to solve five different instances of the space robot pathfinding problem formulated as a VND MO problem where two objectives are considered: (1) the minimal distance of the selected path, and (2) the minimal energy cost (expressed as the number of turning points). VNDMOPSO shows at least comparable or better convergence on the benchmark problems and significantly better convergence properties on the VND pathfinding problems compared with other VND MO algorithms.

**Keywords:** space robot; pathfinding; heuristic algorithm; particle swarm optimization; variable number of dimensions; metameric optimization



## 1. Introduction

Algorithms for finding the optimal path are used in our lives on a daily basis. Fast and accurate methods for finding the optimal route can make our lives more pleasant, e.g., when searching the optimal route using public transport [1], saving a huge amount of resources in cases of a proper delivery path planning [2], or even saving lives in cases of emergency vehicle routing [3].

The minimal distance of the path connecting the starting and target points is usually the first choice for the objective function used by the algorithms for optimal planning on the surface of the earth. Other objectives such as the shortest traveling time, the least power consumption during the travel, or the minimal risks are used by specific applications. Nevertheless, all the mentioned objectives are valid at once for the space robot path planning. Consider, for example, the surface of the moon, where various robot rovers must avoid surface irregularities, whether craters or, conversely, outcrops [4]. Power consumption and route reliability have at least the same importance as the length of the route itself in the case of the space robot path planning. This is due to the unavailability of replacement sources and the very limited (and costly) or complete impossibility of repairing the device in the event of a collision with an obstacle outside the Earth. The conventional graph-based algorithms including the famous Dijkstra's [5] and A-star algorithms [6] follow only one objective during the optimal path search. All decisions are made based on weights of graph node connections that express one of the considered path parameters (the shortest distance, the least power consumption, etc.) or their linear combination where the weights of individual components have to be set a priori.

Nevertheless, the space robot pathfinding problem (SRPP) optimization has to be considered as a multi-objective optimization problem (MOOP) [7] to prevent the loss of

generality of the solution. The set of compromise solutions (the so-called Pareto front) as a result of the multi-objective optimization problem gives much more insight into the conflicting objectives of the solved problem. The final solution can be then selected from the Pareto front by an expert or an automated system considering the gained knowledge about the problem under solution.

Several authors have considered the SRPP as a multi-objective problem. For example, Ajeil et al. consider two objectives during the path planning: (1) the total path's distance, and (2) the path's smoothness in [8]. Nevertheless, they combine the two objectives into a single weighted function, with weights being selected a priori. Another drawback of their approach is that they select the total number of the turning (or way) points (TP) between the start and target position a priori. This drawback can be overcome by employing the so-called grid-based solution, as in [9], where the solution domain is first divided into a set of regular subdomains (usually of a rectangular shape). Then, the subdomains occupied by obstacles cannot become the TPs while the empty subdomains can serve as the TPs. The decision space vector for optimization purposes is then a stream of subdomain addresses and its length has to be either fixed [10,11] or some additional algorithm [9,12] has to control the addition or deletion of TPs. The grid-based methods are relatively easy to implement. Nevertheless, they suffer from several drawbacks: the obstacles are not represented precisely in cases of a very coarse subdomain tessellation, or the number of subdomains reaches very high values, which leads to an unwanted explosion of the dimensions of the decision space vector [13].

The drawbacks of the aforementioned SRPP formulations as either single- or multi-objective problems with fixed dimensions turn our attention to the class of so-called variable number of dimensions (VND) algorithms. Some authors refer to these algorithms as metameric. The VND problems have in common that their solution is built using more structurally similar segments and the optimal number of those segments is usually a priori unknown. The family of metameric problems contains a large variety of problems, starting from a wind farm turbine placement [14], composite laminate metamaterials design [15], and neural network architecture design [16], to biomedical inverse imaging [17]. A comprehensive review of VND problems can be found in [18]. Pure VND algorithms face two tasks during the optimization process: (1) they have to find the optimal dimension of the final solution (i.e., number of segments), and (2) they have to find the optimal values of the individual decision space variables (DSV) [19]. Clearly, the SRPP problem belongs to the multi-objective VND problems because the optimal number of turning points depends on the configuration of the obstacles and more than one objective should be considered during the path selection.

Most of the initial works regarding VND optimization focused on modifications of genetic algorithms [20,21] because of their natural ability to represent the grid-based problems with the binary DSVs. Then, the attention moved to less limited evolutionary optimization algorithms (EOA) such as differential evolution (DE) [22], or heuristic algorithms (HA) such as particle swarm optimization (PSO) [23,24]. Single-objective algorithms based on PSO were proposed in [19,25]. Although there are quite few studies dealing with a single-objective VNDPSO, e.g., [26–29], studies dealing with multi-objective VNDPSO-based algorithms are very rare. The so-called social class MOPSO algorithm (SCMOPSO) was introduced in [30] to solve the VND MO problem of wireless sensor network design. In SCMOPSO, the particles are clustered according to their dimension to different classes. Then, the particles move to another class (change their dimension) after a certain number of iterations in which they do not improve. However, the exchange of information between particles is limited only to members of one class. Authors in [31] proposed the multi-objective variable length PSO (VLMOPSO). VLMOPSO uses an additional binary variable that enables or disables any DSV (the metameric segment to be more precise), which deteriorates the convergence of the algorithm since the decision space dimension is enlarged significantly. In [32], we proposed the variable number of dimensions generalized differential evolution (VNDGDE3)—a VND extension based on the multi-objective

GDE3 algorithm [33]. In VNDGDE3, a large portion of the decision space variables for the crossover operation is selected randomly. This behavior results in the generation of infeasible DSVs (i.e., paths going through obstacles in the case of SRPP).

Promising results of VNDPSO on single-objective problems such as printed circuit board decoupling [34], microwave imaging [17], or design of an antenna array [35] lead us to the idea of extending it to solve multi-objective problems as well. Therefore, the contributions of this paper can be summarized as follows. A novel variable number of dimension multi-objective optimization (VNDMOPSO) algorithm is proposed. Its convergence properties are assessed on a set of VND benchmark problems. The convergence of VNDMOPSO is compared with other available VND algorithms, namely VNDGDE3 [32] and VLMOPSO [31]. The VNDMOPSO is then applied to solve various instances of the multi-objective space robot pathfinding problem where both distance and number of turning points should be minimized.

The rest of the paper is organized as follows. The space robot path planning problem is formulated as the VND two-objective problem in Section 2. Section 3 describes the proposed VNDMOPSO algorithm. Then, Section 4 contains the discussion of obtained results for the VNDMOPSO algorithm and compares them with results obtained by the reference algorithms. The influence of control parameters of VNDMOPSO is assessed here, also. Finally, Section 5 reviews the main conclusions of the paper.

## 2. Problem Definition

The space robot pathfinding problem can be viewed as a search for the set of optimal turning points leading the robot safely from its starting location, $x_S$, to the target location, $x_T$. These TPs have to be located somewhere in the feasible decision space, $\Omega \subset \mathbb{R}^2$. The bounding box surrounding all the obstacles located in the area of interest serves usually as $\Omega$. The situation is illustrated in Figure 1.

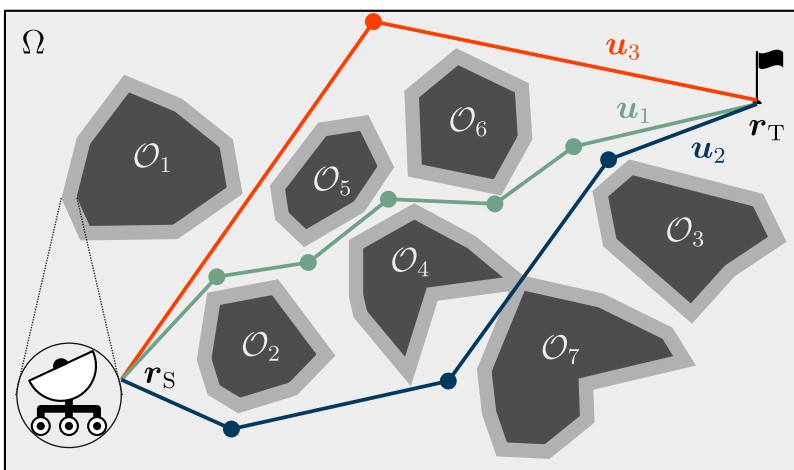

**Figure 1.** The illustration of the space robot path problem. Three feasible paths from the starting point, $r_S$, to the target point, $r_T$, form the Pareto front of the multi-objective SRPP.

First, we have to clarify some assumptions that are valid throughout the whole paper:

**Assumption 1.** *The obstacles are represented by polygons. The n-th polygon is then a set of 2D nodes, $\mathcal{O}_n$, ordered in a counter-clockwise order.*

**Assumption 2.** *The mobile robot has a physical body, which is circumferenced by a circle of radius r. All the obstacles are then enlarged by r (see shaded borders in Figure 1) so that the robot can be viewed as a single point.*

**Assumption 3.** *The obstacles are static and their position is known.*

**Assumption 4.** *The robot moves with a constant speed within* $\Omega$. *The robot can change the direction of the movement at any time omnidirectionally.*

The problem of the robot's path through the space containing several obstacles of different polygonal shapes is then formulated as the VND MOOP as follows:

$$
\begin{aligned}
\min_{\boldsymbol{u}} \quad & f_1(\boldsymbol{u}), f_2(\boldsymbol{u}) \\
\text{s.t.} \quad & \boldsymbol{u} \in \Gamma \subset \mathbb{R}^{N_\mathrm{d}}, \\
& N_\mathrm{d} \in \mathcal{N} = \{2, 4, 6, \ldots, N_\mathrm{max}\}.
\end{aligned} \tag{1}
$$

Here, symbol $\boldsymbol{u}$ denotes a decision space vector whose feasible space, $\Gamma$, is a subspace of $\mathbb{R}^{N_\mathrm{d}}$. All possible values, $N_\mathrm{d}$, generate a list of feasible dimensions ($N$) of the VND problem formulation. The symbol $N_\mathrm{max}$ denotes the largest possible dimension. In this formulation of SPRR, $\boldsymbol{u}$ is formed by individual turning points, $\boldsymbol{r}_i$, on the way from $\boldsymbol{r}_\mathcal{S}$ to $\boldsymbol{r}_\mathcal{T}$. The location of the $i$-th TP is specified by a 2D vector, $\boldsymbol{r}_i = x_i, y_i$. Thus, the DSV is a collection of TPs:

$$
\boldsymbol{u} = \left\{ \boldsymbol{r}_1, \boldsymbol{r}_2, \ldots, \boldsymbol{r}_{N_\mathrm{d}/2} \right\}. \tag{2}
$$

The path through a maze can be then constructed as a collection, $\boldsymbol{p} = \{\boldsymbol{r}_\mathcal{S}, \boldsymbol{u}, \boldsymbol{r}_\mathcal{T}\}$.

Referring to (1), the symbols $f_1$ and $f_2$ stand for the two considered objective functions. The first objective function minimizes the traveled Euclidean distance of the path:

$$
f_1(\boldsymbol{u}) = \sum_{i=1}^{N_\mathrm{d}/2+1} d(\boldsymbol{p}_i, \boldsymbol{p}_{i+1}), \tag{3}
$$

where $\boldsymbol{p}_i$ is the $i$-th location from the path and $d(\cdot, \cdot)$ denotes the Euclidean distance between two locations. The second objective function takes the complexity of the path into consideration:

$$
f_2(\boldsymbol{u}) = ||\boldsymbol{u}||/2, \tag{4}
$$

where $|| \cdot ||$ denotes the length of the vector. Thus, $f_2$ is a discrete objective function and minimizes the number of TPs. There are three candidate solutions: $\boldsymbol{u}_1$ (green curve), $\boldsymbol{u}_2$ (blue), and $\boldsymbol{u}_3$ (red) in Figure 1. These three candidate solutions have 5, 3, and 1 TPs, respectively, which results in DSV dimensions $N_\mathrm{d} = 10, 6$, and 2, respectively. It should be noted that any other objectives could be added to the problem formulation (1), e.g., the maximization of a minimal distance of the considered path towards any of the obstacles. For the sake of clarity of the shown results, we stick to the two-objective formulation in this paper.

## 3. Optimization Methods

This section reviews the optimization methods whose results are further discussed in the paper. First, we introduce a detailed description of the VNDMOPSO algorithm. Then, we briefly review the state-of-the-art VND algorithms VLMOPSO and VNDGDE3 used as reference methods in this study.

### 3.1. Variable Number of Dimensions MOPSO

#### 3.1.1. Conventional MOPSO

The algorithm VNDMOPSO follows the principles of standard single- and multi-objective PSO [23,36]. Therefore, we briefly review the main steps of the MOPSO algorithm [36] that serves as a starting point for the development of the VNDMOPSO algorithm. The MOPSO starts as any other HA with a random generation of particles:

$$\boldsymbol{u}_p = \boldsymbol{u}_{\min} + \boldsymbol{r} \otimes (\boldsymbol{u}_{\max} - \boldsymbol{u}_{\min}), \tag{5}$$

where $p$ stands for the index of an agent from a total number of $N_A$ agents, $\otimes$ denotes the element-wise multiplication, and $\boldsymbol{r}$ is the vector of random values from interval $\langle 0; 1.0 \rangle$ with a uniform probability of distribution. The decision space is limited by the lower and upper bounds, $\boldsymbol{u}_{\min}$ and $\boldsymbol{u}_{\max}$, respectively. Then, the positions, $\boldsymbol{u}$, of particles are iteratively updated based on the formula:

$$\boldsymbol{u}_p(i) = \boldsymbol{u}_p(i-1) + \Delta_t \boldsymbol{v}_p(i), \tag{6}$$

where $\Delta_t$ is the time step ($\Delta_t = 1\,\mathrm{s}$ is used in the vast majority of the studies) and $\boldsymbol{v}_p(i)$ stands for the current velocity vector that reads:

$$\boldsymbol{v}_p(i) = w\boldsymbol{v}_p(i-1) + c_1 r_1 \big[\boldsymbol{b}_p - \boldsymbol{u}_p(i-1)\big] + c_2 r_2 \Big[\boldsymbol{g}_p - \boldsymbol{u}_p(i-1)\Big]. \tag{7}$$

Here, symbols $w$, $c_1$, and $c_2$ stand for the user-defined controlling parameters of the algorithm, in particular the inertia weight, the cognitive learning factor, and the social learning factor, respectively. Then, symbols $\boldsymbol{b}_p$, and $\boldsymbol{g}_p$ stand for the personal and global best positions assigned for the $p$-th particle, respectively. Finally, symbols $r_1$ and $r_2$ are random values from interval $\langle 0; 1.0 \rangle$ that scale the vectors from previous positions of particles to personal and global best, respectively. The total movement of the particle expressed in terms of (6) and (7) is the result of the action of three forces: (1) it tends to keep the previous direction of the movement (scaled by the inertia weight $w$); (2) it is attracted towards the location of the personal best $\boldsymbol{b}$ (scaled by $c_1 r_1$); and (3) it is attracted towards the global best $\boldsymbol{g}$.

In single-objective PSO, every particle has its own $\boldsymbol{b}_p$ and the same $\boldsymbol{g}$ is used for all the particles. The personal and global best are assigned according to the value of the objective function $f$. In MOPSO, a set of objective function values has to be considered when selecting the $\boldsymbol{b}_p$ and $\boldsymbol{g}_p$. Moreover, a different global best vector, $\boldsymbol{g}_p$, is assigned to every particle. The initial particle's position $\boldsymbol{u}_p(1)$ is used for $\boldsymbol{b}_p$ in the first iteration. Then, the personal best is updated to $\boldsymbol{u}_p(i)$ if it dominates $\boldsymbol{b}_p$ (i.e., it is better or equal in all watched objectives).

The global best values, $\boldsymbol{g}_p$, are assigned from the so-called external archive, $\mathcal{E}$, that stores all non-dominated solutions found so far [36]. It is updated at the end of every iteration, $i$. The size of the external archive, $\mathcal{E}$, is limited to $N_A$ to avoid the exponential growth of the algorithm's computational complexity. The effective method for pruning of $\mathcal{E}$ [37] is used for the selection of members of $\mathcal{E}$. The mechanism of the global best selection combines two approaches: (1) random assignment; and (2) assignment based on the Euclidean distance. The global best assignment is shown in the form of pseudocode in Algorithm 1. A random value, $r$, is generated and compared with a value of the user-defined parameter probability of random global best, $0 \le r_g \le 1.0$. If $r < r_g$, then a random member of the external archive, $\mathcal{E}$, is selected for the $p$–th particle to serve as the $\boldsymbol{g}_p$. Contrarily, if $r \ge r_g$, then the closest member from $\mathcal{E}$ is selected as $\boldsymbol{g}_p$. When $r_g = 1.0$, the global best positions are selected randomly from $\mathcal{E}$ for all particles. Combining these two approaches is crucial to avoid the MOPSO getting stuck in the local optimum (local Pareto front) and to enable the best possible convergence.

---

**Algorithm 1:** Global best selection algorithm used in MOPSO

---

    **Input**   : Probability of random global best $r_{\mathrm{g}}$, set of DSVs $\mathcal{U}$, external archive $\mathcal{E}$
    **Output**: Set of global best positions $\mathcal{G}$

  1: Find $n = \text{length } \mathcal{U}$
  2: Find $m = \text{length } \mathcal{E}$
  3: **for** $p = 1$, $p \le n$, $p + +$ **do**
  4:      Generate random value $r \in \langle 0; 1.0 \rangle$
  5:      **if** $r < r_{\mathrm{g}}$ **then**
  6:          Select random index $i$ from list $1, 2, \ldots, m$
  7:      **else**
  8:          Find Eucl. distances $\boldsymbol{d}_p$ from $\mathcal{U}[p]$ to all members of $\mathcal{E}$
  9:          Find index $i$ of minimal Eucl. distance $\boldsymbol{d}_p$
10:      **end if**
11:      Set $\mathcal{G}[p] = \mathcal{E}[i]$
12: **end for**
13: Return $\mathcal{G}$

---

After the position update (6), the particles' locations can reside outside the limits of the feasible decision space, $\Gamma$. Then, the so-called boundary conditions can be applied to make the particles feasible again [36]. The possible boundary conditions are: (1) reflecting (the particle is reflected back to $\Gamma$ by the violated limit), (2) absorbing (the particle resides on the violated limit), and (3) invisible (the particle is left outside of $\Gamma$ but its objective function values are worsened artificially). At the end of every iteration, the new positions of particles are evaluated using the objective functions. Then, the external archive is updated. The iterative process of MOPSO continues until the termination condition is met. Usually, a combination of several termination conditions is used, e.g., maximum number of iterations, $N_{\mathrm{I}}$, or limits for the values of objective functions are reached, etc.

### 3.1.2. VNDMOPSO

The MOPSO algorithm as described in the previous subsection is used as the general framework of the novel VNDMOPSO algorithm. The MOPSO algorithm also needs only few significant changes in order to solve VND–formulated problems. First, the particle's dimension, $N_p$, in the initial iteration $i = 1$ is selected randomly for every $p$–th particle. All dimensions from the list of possible dimensions, $\mathcal{N}$, should be selected with a uniform probability. The number of particles with individual dimensions should be approximately equal to $N_{\mathrm{A}}/||\mathcal{N}||$. The particles' dimensions are distributed so that there are no or minimal differences between sizes of dimensional clusters, i.e., sets of particles with the same dimension. If $N_{\mathrm{A}} \ge ||\mathcal{N}||$, it should be ensured that at least one particle is in every dimensional cluster from the list $\mathcal{N}$.

Next, we need to enable the DSVs $\boldsymbol{u}$, manipulated by VNDMOPSO, to have different dimensions. Therefore, the four vectors, $\boldsymbol{g}_p$, $\boldsymbol{b}_p$, $\boldsymbol{v}_p$, and $\boldsymbol{u}_p$, used in (7), can be of three different lengths:

1.      $N_{\mathrm{g}}$: size of the global best, $\boldsymbol{g}$;
2.      $N_{\mathrm{b}}$: size of the personal best, $\boldsymbol{b}$;
3.      $N_{\mathrm{u}}$: the previous dimension of the agent $\boldsymbol{u}(i-1)$ (and its velocity $\boldsymbol{v}(i-1)$, also).

The new dimension, $N_p$, of the $p$-th particle is determined based on the simple approach presented in Algorithm 2. The new dimension is selected on a random basis but taking into account user-defined probabilities [19]:

1.      $p_{\mathrm{g}}$: probability of global best dimension, $N_{\mathrm{g}}$;
2.      $p_{\mathrm{b}}$: probability of personal best dimension, $N_{\mathrm{b}}$.

The probabilities are set so that $(p_g + p_b) \leq 1.0$, as shown in Figure 2a. The lower the values of $p_g$ and $p_b$ are, the higher the diversity of dimensions of particles is. In an extreme case when $p_g + p_b = 0$, all the particles remain in their initial randomly selected dimension. On the other hand, the dimensional diversity is quickly lost when $p_g + p_b = 1.0$. In that case, the particles are forced to change their dimension to the dimension of non-dominated DSVs that are found in the early iterations of the optimization and stored in $\mathcal{E}$.

---

**Algorithm 2:** New dimension selection in VNDMOPSO

---

**Input**   : Probabilities $\{p_g, p_b\}$, dimensions $N_g, N_b, N_u$ for $p$–th agent
**Output**: New dimension $N_p$ of $p$–th agent
  1: Generate random value $r \in \langle 0; 1.0 \rangle$
  2: **if** $r < p_g$ **then**
  3:    $N_p = N_g$
  4: **else if** $p_g \leq r < p_g + p_b$ **then**
  5:    $N_p = N_b$
  6: **else**
  7:    $N_p = N_u$
  8: **end if**
  9: Return $N_p$

---

After the new dimension, $N_p$, of the $p$-th particle is selected using Algorithm 2, the potentially different dimensions of vectors $\boldsymbol{g}_p$, $\boldsymbol{b}_p$, $\boldsymbol{v}_p$, and $\boldsymbol{u}_p$ have to be balanced. The process of balancing the dimensions of the vectors is shown in Figure 2b. The vectors whose dimensions are higher than the selected one are trimmed so that their sizes are equal to $N_p$. On the other hand, if any vector has fewer elements than $N_p$, the missing elements are taken from the other vector, having at least $N_p$ elements. Please note that the priority is decreasing from the $\boldsymbol{g}_p$, over $\boldsymbol{b}_p$, to $\boldsymbol{v}_p$, so that the convergence is strengthened. The global best, $\boldsymbol{g}_p$, and personal best, $\boldsymbol{b}_p$, contain non-dominated vectors.

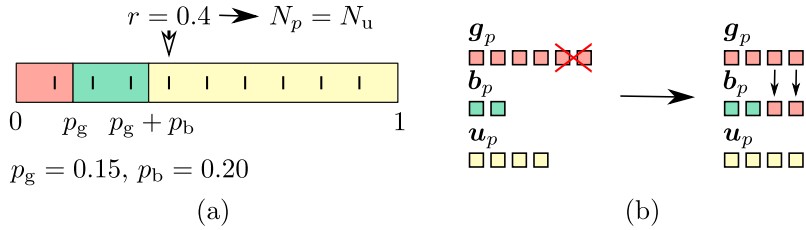

**Figure 2.** The VNDMOPSO DSVs manipulation for the velocity update formula: (**a**) the selection of new dimension, $N_p$, of the $p$-th particle, (**b**) corresponding update of vectors global best, $\boldsymbol{g}_p$, personal best, $\boldsymbol{b}_p$, and previous position, $\boldsymbol{u}_p$.

The general pseudocode of the VNDMOPSO is summarized in Algorithm 3. The VNDMOPSO needs inputs in the form of (1) problem definition (the definition of objective functions, $f$, the list of feasible dimensions, $\mathcal{N}$, and the limits of the decision space, $\Omega$); and (2) set of user-defined control parameters, $\mathcal{S}$. The list of control parameters and their recommended values based on the results of different benchmark studies are summarized in Table 1.

---

**Algorithm 3:** VNDMOPSO general pseudocode

---

**Input**　:Set of user-defined controling parameters $\mathcal{S}$, objective functions $\boldsymbol{f}$, list of feasible dimensions $\mathcal{N}$, decision space limits $\Gamma$

**Output**:Non-dominated set $\mathcal{U}$

1: Generate random dimension for every particle in the swarm.
2: Generate random initial positions $\boldsymbol{u}(i = 1)$ and velocities $\boldsymbol{v}(1)$.
3: Evaluate objective function values for all particles $\boldsymbol{f}(\boldsymbol{u})$.
4: Create extarnal archive $\mathcal{E}$.
5: Assign personal/global best positions (see Algorithm 1).
6: **while** Stop condition **do**
7: 　**for** Every particle $\boldsymbol{u}_p$ **do**
8: 　　Find current dimension $N_p$ using Algorithm 2.
9: 　　Create copies of previous $(i - 1)$ DSVs $\boldsymbol{g}_p$, $\boldsymbol{b}_p$, and $\boldsymbol{u}_p$ $(\boldsymbol{v}_p)$ with a dimension $N_p$.
10: 　　Update the velocity $\boldsymbol{v}_p(i)$, and position $\boldsymbol{u}_p(i)$ using (6), and (7).
11: 　　**if** $\boldsymbol{u}_p(i) \notin \Gamma$ **then**
12: 　　　Apply boundary condition on $\boldsymbol{u}_p(i)$.
13: 　　**end if**
14: 　　Evaluate objective function values $\boldsymbol{f}\big(\boldsymbol{u}_p(i)\big)$.
15: 　　Update the personal best $\boldsymbol{b}_p(i)$.
16: 　**end for**
17: 　Update external archive $\mathcal{E}$.
18: 　Update global best positions using Algorithm 1.
19: **end while**
20: Return non-dominated set $\mathcal{U} = \mathcal{E}$.

---

**Table 1.** The list of VNDMOPSO controlling parameters and their recommended values.

| Symbol | Explanation | Value Range |
|:---:|:---:|:---:|
| $N_A$ | Number of particles (agents) | $\langle 2\|\|\mathcal{N}\|\|; 100\rangle$ |
| $N_I$ | Maximal number of iterations | $\langle 20; 100\rangle$ |
| $w$ | Inertia weight | $\langle 0.40; 0.90\rangle$ |
| $c_1$ | Cognitive learning factor | $\langle 0.90; 1.90\rangle$ |
| $c_2$ | Social learning factor | $\langle 0.90; 1.90\rangle$ |
| $p_g$ | Probability of adapting to dimension of global best | $\langle 0.01; 0.30\rangle$ |
| $p_b$ | Probability of adapting to dimension of personal best | $\langle 0.01; 2p_g\rangle$ |

### *3.2. Reference Methods*

The two VND multi-objective algorithms available in the open literature, namely variable length MOPSO and VNDGDE3, are briefly reviewed in the following subsections. They are used as comparative algorithms in this study.

#### 3.2.1. VLMOPSO

Algorithm VLMOPSO was introduced by Mukhopadhyay and Mandal in [31]. It works on the principle of the DSV extension. The DSV has two parts: the model variables (initial elements of the DSV, i.e., degrees of freedom of the designed structure), and the so-called padding variables. Using this approach, all DSVs are working with the maximal dimension, $N_{max}$, for the model variables. The padding variables are in a binary form. If the padding variable equals 1 then the corresponding part of the model variable is taken into account, and vice versa. The number of padding variables, $N_P$, is determined by the number of metameric (clustered) variables. In this study, the number of padding variables is the number of turning points, i.e., $N_{max}/2$. Adding padding variables to $\boldsymbol{u}$ makes the optimization much more complex following the curse of dimensionality: the optimizer has

to search for the optimal combination of more variables that are necessary to specify the designed structure or system.

### 3.2.2. VNDGDE3

The VND extension of a well-known GDE3 algorithm was introduced in [32] by Marek and Kadlec. In GDE3 [33], the so-called trial vector is created for every agent in the current generation of agents (DSVs). It is composed based on three other randomly selected agents and the previous agent's position. Then, the crossover vector is accepted, if it dominates the agent's previous position. Otherwise, the agent stays at its previous position.

The dimension of the trial vector in VNDGDE3 is determined using a similar approach as in this study. The user-defined parameter called probability of dimension transition, $p_{DT}$, controls the change of dimension of every agent. Similarly, as in Algorithm 2, a random value, $r \in \langle 0; 1.0 \rangle$, is generated. If $r \leq p_{DT}$ then the agent keeps its current dimension, $N_u$. Otherwise, the agent's new dimension is selected according to one of the three randomly selected agents to produce the trial vector. It should be noted that they all have an equal chance of being selected $((1 - p_{DT})/3)$.

### 3.2.3. SCMOPSO

The SCMOPSO algorithm was designed to solve the VND MO wireless network sensor deployment by Jubair et al. in [30]. It works with a population of particles that are clustered into different so-called classes based on the dimension of individual particles. The movement of a $p$-th particle is driven by the so-called exemplar. The exemplar is selected from the other particles that are in the same class as particle $\boldsymbol{u}_p$. The particles with better values of the objective function are favored to be selected as exemplars. If a particle does not improve for a pre-defined number of iterations, it can move to another class. However, there is a restriction on the minimal size of the class. The new class is selected based on the probability density function that is constructed over the values for one randomly selected objective function for the best particles from different classes. Finally, SCMOPSO adds the mutation operation to the PSO algorithm. A randomly chosen variable of a particle can be mutated with a certain probability, which means that the variable will be generated randomly in the feasible decision space.

## 4. Results and Discussion

The novel VNDMOPSO algorithm is validated here to prove its ability to solve as many different VND MO problems as possible. First, it solves various benchmark problems and the results are compared with two state-of-the-art VND MO algorithms, namely VLMOPSO and VNDGDE3. Then, VNDMOPSO is applied to five pathfinding problem instances. The results are compared with results obtained for VLMOPSO, VNDGDE3, and SCMOPSO algorithms. All the algorithms except for SCMOPSO are implemented using a standalone MATLAB-based toolbox called Fast Optimization ProcedureS (FOPS) [38] that is available online. FOPS is an in-house code developed and maintained at Brno University of Technology. All the comparative tests are set so that the compared algorithms use the same number of agents, $N_A$, and iterations $N_I$ to keep the comparisons fair. If not otherwise stated, the controlling parameters of VNDMOPSO are set as summarized in Table 2. The other algorithms use their default settings as described in [38] and in [30] (for the SCMOPSO). Next, the influence of the VNDMOPSO control parameters is evaluated using the benchmark problems.

**Table 2.** Default values of VNDMOPSO controlling parameters.

| Symbol | Explanation | Value Range |
| --- | --- | --- |
| $w$ | Inertia weight | Decreasing from 0.9 to 0.4 |
| $c_1$ | Cognitive learning factor | 1.5 |
| $c_2$ | Social learning factor | 1.5 |
| $p_g$ | Probability of adapting to dimension of global best | 0.02 |
| $p_b$ | Probability of adapting to dimension of personal best | 0.04 |

*4.1. Benchmark Problems*

The results of the VNDMOPSO are first validated using the available benchmark problems with known Pareto fronts and corresponding Pareto-optimal sets. There is a large set of benchmark metrics to assess the quality of the found solution of a multi-objective benchmark problem. Some of them are focused on how close the found non-dominated solutions (members of $\mathcal{E}$) are to the true Pareto front members, $\mathcal{PF}$, in the objective space, e.g., the generational distance [39]. Other metrics, e.g., the spread [7], express the uniformness of the found non-dominated set over the true $\mathcal{PF}$. The commonly accepted way to express both of these requirements for a multi-objective algorithm using a single numerical value is a metric called hypervolume (HV), introduced in [40]. It determines the portion of the objective space volume that is dominated by the found set of non-dominated solutions, $\mathcal{E}$. The hypervolume is computed as the union of volumes between the individual members of $\mathcal{E}$ and the reference point [41]:

$$\mathrm{HV}^{\mathcal{E}} = \cup_{i=1}^{||\mathcal{E}||} V(\boldsymbol{e}_i).\tag{8}$$

Here, $V(\boldsymbol{e}_i)$ denotes the volume dominated by the $i$-th member of the non-dominated set, $\mathcal{E}$, i.e., the volume of a hyper-cube with opposite points, $\boldsymbol{e}_i$, and the reference point, $\boldsymbol{p}$. The so-called nadir point (the collection of the worst values of individual objectives from individual extreme points of the true $\mathcal{PF}$ [7]) is usually selected as the reference point. The larger the HV is, the better the result of the MO algorithm has been found.

The only disadvantage of HV is that it's value depends on the selection of the reference point. Therefore, the distance hypervolume metric was introduced to overcome this problem:

$$\mathrm{dHV} = \mathrm{HV}^{\mathcal{PF}} - \mathrm{HV}^{\mathcal{E}}.\tag{9}$$

It is computed as the difference between the hypervolume dominated by the true Pareto front, $\mathrm{HV}^{\mathcal{PF}}$, and by the found one, $\mathrm{HV}^{\mathcal{E}}$. Thanks to this method of calculation, the influence of the choice of the reference point is neglected.

The suite of benchmark problems used in this study contains 21 VND multi-objective problems whose definition was introduced in [32,42]. These benchmark problems are based on families of well-known test suites such as DTLZ [43], ZDT [44], LI [42], and LZ [45]. The modified benchmark problems have a feature that different parts of the Pareto front can be occupied by DSVs with different dimensions. The full definition of the benchmark problems is beyond the scope of this paper. However, the method to determine the optimal dimension over the Pareto front and definitions of all the benchmark problems are summarized in the supplementary material S1.

We compare the convergence properties of the novel VNDMOPSO algorithm with VLMOPSO and VNDGDE3 on the full benchmark suite using three different dimension settings to see how the convergence of the algorithms scales with the dimensionality of the problem. The dimensionality settings are summarized in Table 3.

**Table 3.** Dimensionality settings for the comparative study between VNDMOPSO and reference algorithms VLMOPSO and VNDGDE3.

| Settings | Feasible Dimensions, $\mathcal{N}$ | Optimal Dimensions, $D^{(\text{opt})}$ |
|:---:|:---:|:---:|
| $\mathcal{S}1$ | $3, 4, \ldots, 12$ | $3, 4, 5$ |
| $\mathcal{S}2$ | $3, 4, \ldots, 22$ | $7, 8, 9$ |
| $\mathcal{S}3$ | $3, 4, \ldots, 52$ | $10, 11, 12$ |

The results in form of boxplots of the dHV metric are shown in Figures 3–5. The results from 100 independent runs of individual algorithms were compared using the Wilcoxon's ranked sum test. Please note that the test was performed at the significance level $\alpha = 0.05$. The results of the Wilcoxon's test comparing VNDMOPSO with VNDGDE3 and VNDMOPSO with VLMOPSO are summarized in Table 4.

**Table 4.** Results of Wilcoxon's test (at significance level $\alpha = 0.05$) when comparing the VNDMOPSO algorithm with VNDGDE3 and VLMOPSO for different dimensionality settings: '+' denotes that the VNDMOPSO algorithm is significantly better, '−' denotes that the second algorithm is significantly better, '=' denotes that the difference is not significant. The last row summarizes the number of the test results for the whole benchmark set: VNDMOPSO is better/difference not significant/VNDMOPSO is worse.

| | Settings $\mathcal{S}1$ | | Settings $\mathcal{S}2$ | | Settings $\mathcal{S}3$ | |
|:---|:---:|:---:|:---:|:---:|:---:|:---:|
| **VNDMOPSO vs.** | **VNDGDE3** | **VLMOPSO** | **VNDGDE3** | **VLMOPSO** | **VNDGDE3** | **VLMOPSO** |
| VNDMODTLZ1 | − | + | − | + | − | = |
| VNDMODTLZ2 | + | − | + | − | + | − |
| VNDMODTLZ3 | − | + | − | = | − | = |
| VNDMODTLZ4 | − | − | − | − | − | − |
| VNDMODTLZ5 | + | + | + | − | + | + |
| VNDMODTLZ6 | = | + | + | + | + | − |
| VNDMODTLZ7 | − | − | − | − | − | = |
| VNDMOLI1 | − | + | − | + | − | + |
| VNDMOLZ1 | − | − | + | + | + | + |
| VNDMOLZ2 | − | − | − | + | − | + |
| VNDMOLZ3 | − | + | + | + | + | + |
| VNDMOLZ4 | − | + | + | + | = | + |
| VNDMOLZ5 | = | + | + | + | + | + |
| VNDMOLZ7 | = | = | + | = | + | + |
| VNDMOLZ8 | − | = | + | + | + | + |
| VNDMOLZ9 | − | − | − | + | − | + |
| VNDMOZDT1 | + | − | − | − | − | − |
| VNDMOZDT2 | − | − | − | − | − | − |
| VNDMOZDT3 | + | + | − | − | − | − |
| VNDMOZDT4 | + | − | + | − | + | + |
| VNDMOZDT6 | − | − | − | − | + | − |
| **Overall** | 5 / 3 / 13 | 9 / 2 / 10 | 10 / 0 / 11 | 10 / 2 / 9 | 10 / 1 / 10 | 11 / 3 / 7 |

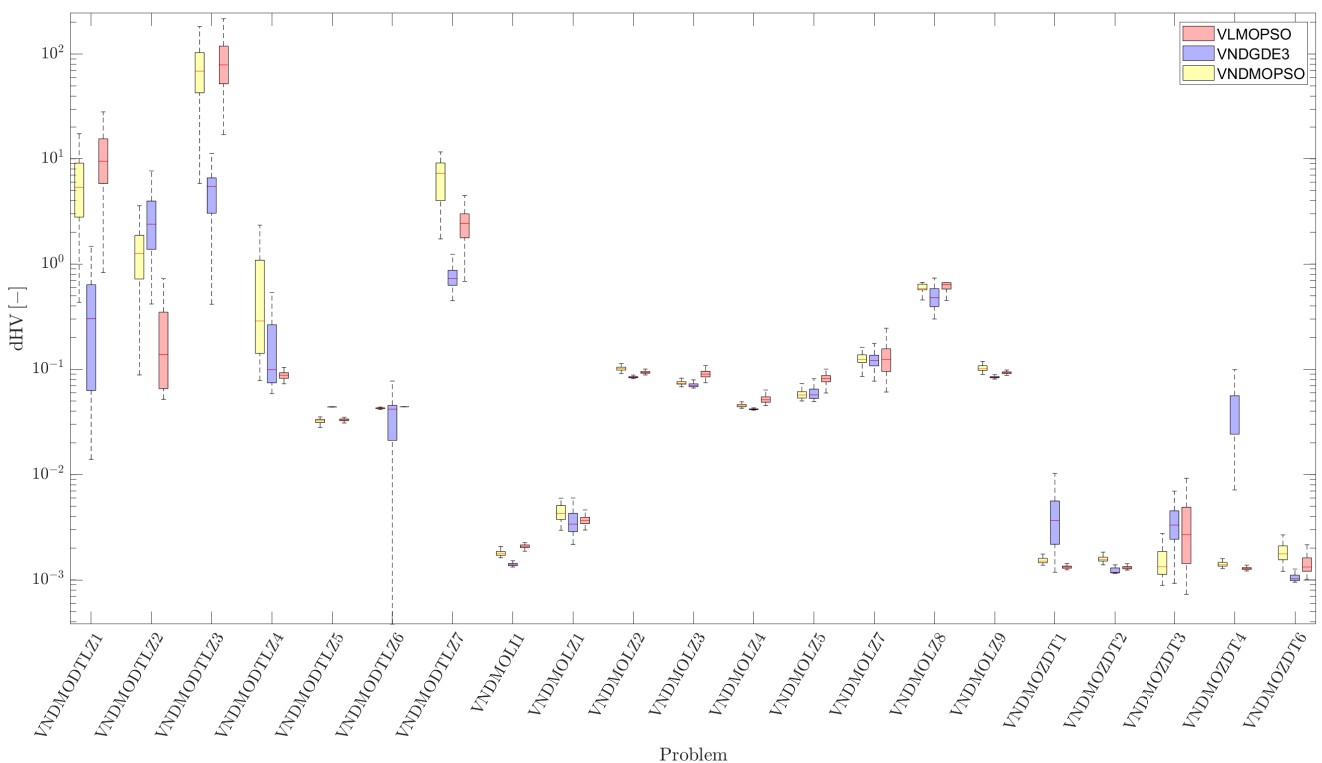

**Figure 3.** Results of dHV metric for algorithms VNDMOPSO, VLMOPSO, and VNDGDE3 for dimensionality settings $\mathcal{S}1$: $N_d \in \{2, 3, \ldots, 12\}$.

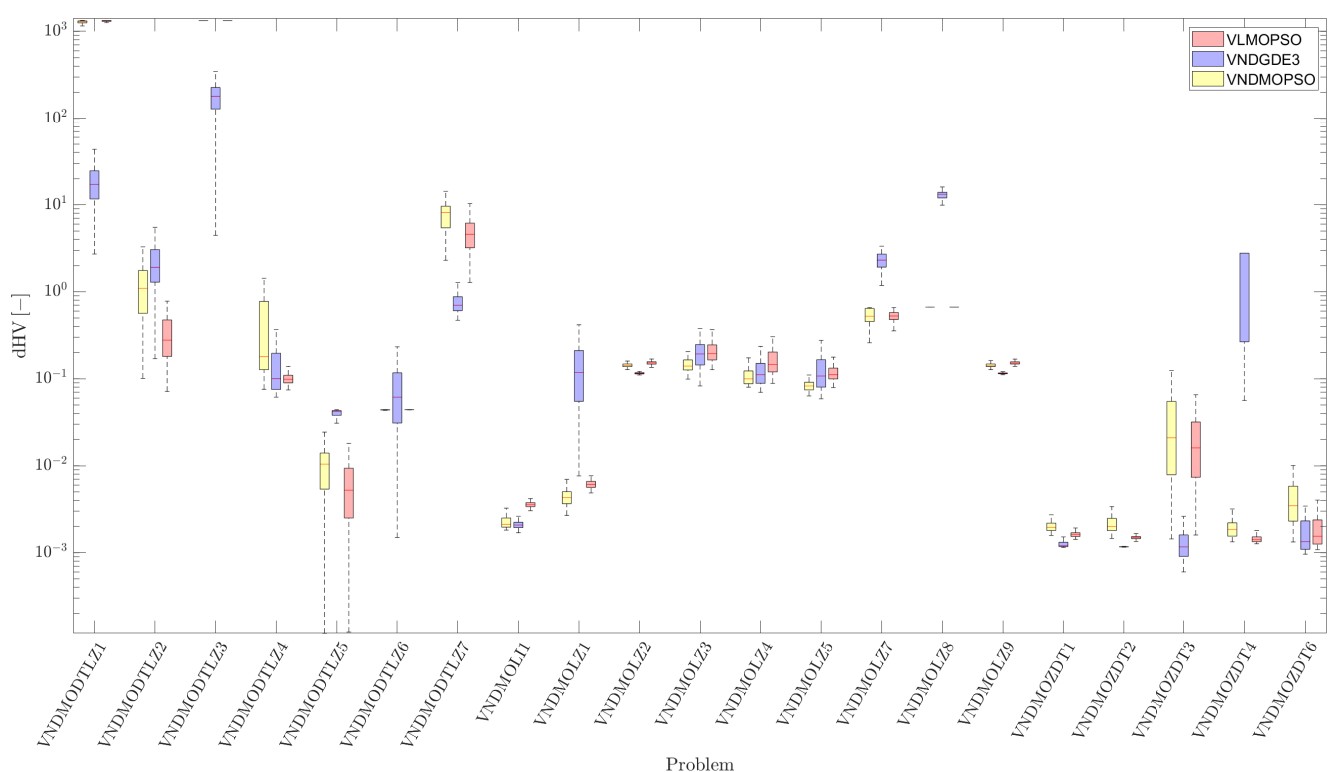

**Figure 4.** Results of dHV metric for algorithms VNDMOPSO, VLMOPSO, and VNDGDE3 for dimensionality settings $\mathcal{S}2$: $N_d \in \{2, 3, \ldots, 22\}$.

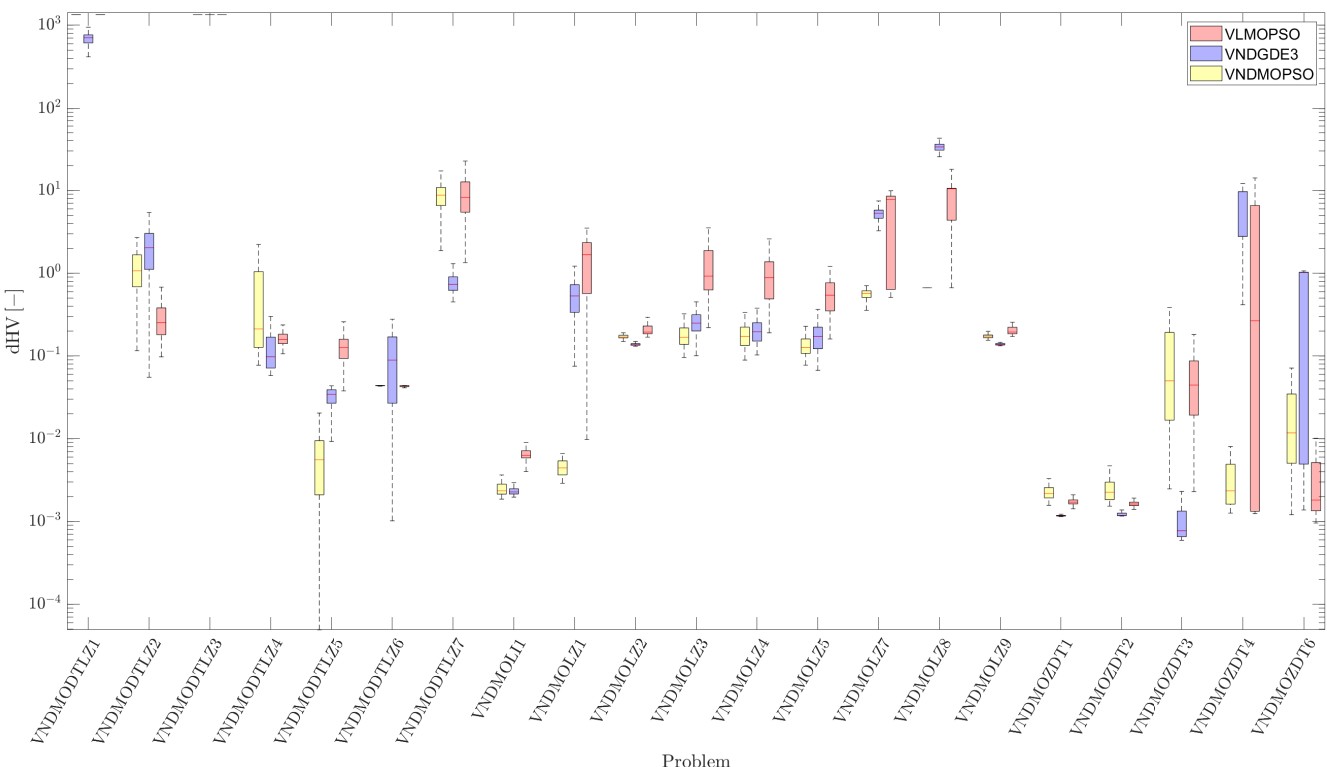

**Figure 5.** Results of dHV metric for algorithms VNDMOPSO, VLMOPSO, and VNDGDE3 for dimensionality settings $\mathcal{S}3$: $N_\mathrm{d} \in \{2, 3, \ldots, 52\}$.

When comparing VNDMOPSO with VNDGDE3, we can see that the performance of VNDMOPSO is inferior for the settings with lower dimensions ($\mathcal{S}1$: $N_\mathrm{max} = 12$) whereas the VNDGDE3 is significantly better for 13 problems while the VNDMOPSO is better for 5 problems only. The results become more comparable for the more-dimensional settings, $\mathcal{S}2$ and $\mathcal{S}3$. For them, VNDMOPSO significantly outperforms VNDGDE3 for at least 10 problems. Next, it is clearly visible that VNDMOPSO achieves at least comparable results to VLMOPSO for settings $\mathcal{S}1$ and outperforms VLMOPSO for more-dimensional settings, $\mathcal{S}2$ and $\mathcal{S}3$. Most importantly, the VNDMOPSO algorithm outperforms the other two algorithms on problems from the LZ benchmark suite (please refer to supplementary material S1, Equations (S22)–(S39)). The nature of these problems is very similar to searching for the correct location of some points in an $N$–dimensional space, which is similar to the main subject of this paper—searching for the locations of the turning points of a space robot. Therefore, VNDMOPSO can be seen as the best candidate to solve the SPRR.

### 4.2. Parameter $p_\mathrm{g}$ Influence

The only new control parameters of VNDMOPSO compared with the initial algorithm MOPSO [36] are the two probabilities of dimensional transition to the dimension of global best, $p_\mathrm{g}$, and personal best, $p_\mathrm{b}$. Their values determine the rate of the change of particle's dimension. Thus, they can balance the exploration vs exploitation dilemma that arises whenever any EOA or HA is used. By the proper setting of $p_\mathrm{g}$ and $p_\mathrm{b}$, we can either prioritize the speed of convergence or, on the contrary, emphasize the certainty of finding a correct solution. By setting values $p_\mathrm{g}$ and $p_\mathrm{b}$ close to zero, the particles can change their dimension only sporadically, which maintains the dimensional diversity of the swarm. On the other hand, the particles start to form clusters of a certain dimension very soon and thus search it much more carefully, with the risk that the global optimum resides in the abandoned dimensions. This behavior can be well documented by the results shown in Figure 6. Here, the sizes of the clusters containing agents with a certain value of TPs, $N_\mathrm{T}$, are shown against the iterations for a problem, $\mathcal{P}4$, having a discrete Pareto front

consisting of solutions from three different dimensions. While in the first case in Figure 6a, the parameters $p_g = 0.05$ and $p_b = 0.10$ are set robustly, the individual particles change their dimension very gradually, which allows a careful search of the individual dimensions and results in a Pareto front with three solutions having three different dimensions. In the second case, shown in Figure 6b, with $p_g = 0.20$ and $p_b = 0.40$ set rather to speed up the convergence of the VNDMOPSO algorithm, the algorithm finds a solution in the form of a two-element Pareto front, where the solution with dimension $N_T = 3$ is missing. This is caused by the fact that a correct combination of a six-element DSV is the most difficult one to find.

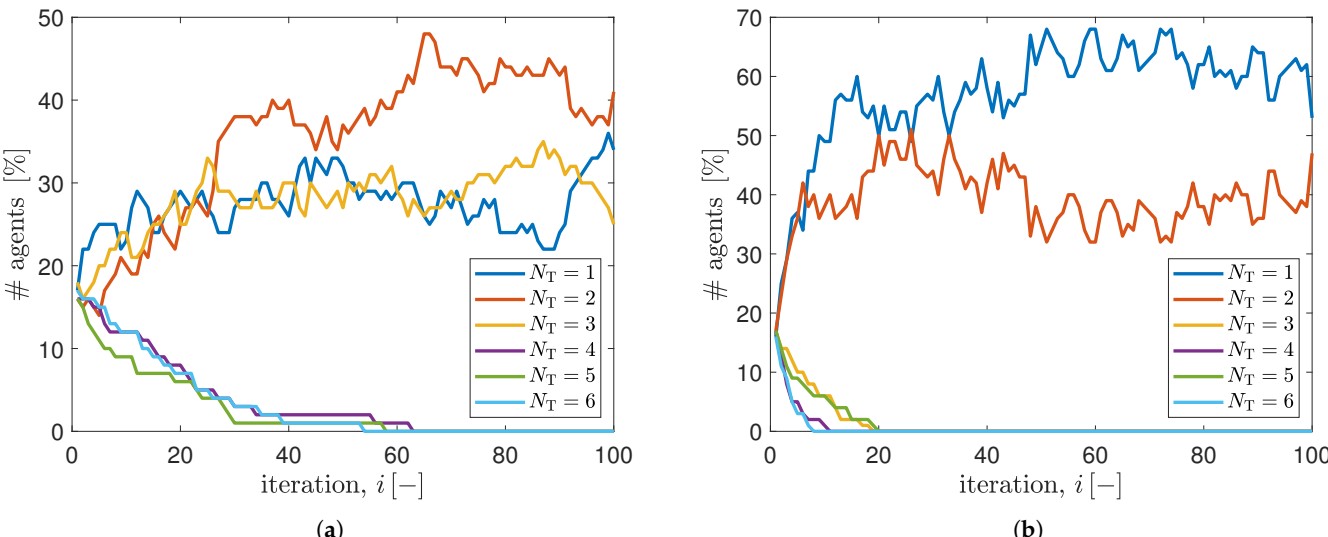

**Figure 6.** Percentage of agents with different dimensions expressed in terms of the number of turning points, $N_T$, against the number of iterations on $\mathcal{P}4$ problem instance for two settings of parameters, $p_g$ and $p_b$: (**a**) $p_g = 0.05$ and $p_b = 0.10$; and (**b**) $p_g = 0.20$ and $p_b = 0.40$.

Based on our experience with the algorithm, it is a good strategy to set $p_b = 2p_g$, as recommended in Table 1. In that case, the dimensional diversity is enhanced, since the personal best dimension, $N_b$, and the particle's dimension, $N_p$, are the same. Moreover, only one parameter ($p_g$) needs to be set by the user.

The influence of the $p_g$ on the quality of the solution found by VNDMOPSO is shown in Figure 7. There, the heat plots of the dHV metric are shown for all the benchmark problems and $p_g$ values ranging from interval $\langle 0.1; 1.0 \rangle$. Please note that the dHV metric values are normalized according to obtained values for every individual benchmark problem so that they can be visualized in a single figure. From the results in Figure 7, it can be seen that, for more complex tasks such as those from the DTLZ family (taking into account the results in Figures 3–5), VNDMOPSO converges better for $p_g > 0.8$. The algorithm converges best for less complex tasks such as ZDT1, ZDT4, and LZ1 for $p_g < 0.2$. Concerning the intended application (robot pathfinding, where we prefer a higher certainty of finding the correct solution over the convergence speed), we recommend a compromise value of $p_g \leq 0.30$.

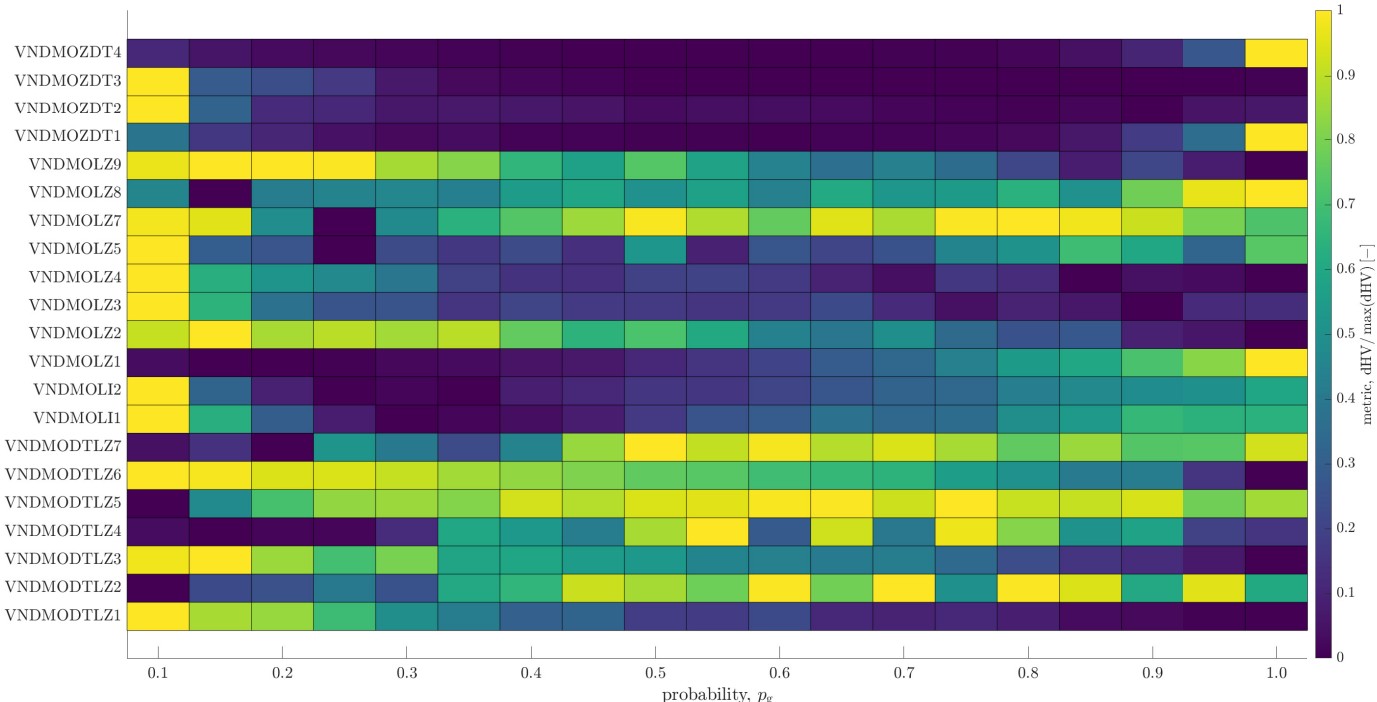

**Figure 7.** Median values of dHV metric versus parameter $p_g$. The median is selected from 100 independent runs of the VNDMOPSO algorithm for 21 benchmark problems with dimensional settings $\mathcal{S}1$.

### 4.3. Pathfinding Problems

In this section, we will apply the VND algorithms to solve four artificial (labeled as $\mathcal{P}1$–$\mathcal{P}4$) and one real-life ($\mathcal{P}5$) pathfinding problem instances. The feasible dimensionalities for all the problems are $\mathcal{N} = \{2, 4, \ldots, 20\}$. Maps of obstacles for the problems are shown in Figure 8. They were designed so that they can be solved with different numbers of TPs. See e.g., problem $\mathcal{P}4$ where the Pareto front constitutes solutions leading to three different paths with one, two, or three TPs (see Figure 8). The real-life problem instance, $\mathcal{P}5$, is then defined based on the photograph of the landing area of the famous Apollo 11 mission reaching the surface of the Moon [46], see Figures 8e,f.

The problem instance, $\mathcal{P}4$, is a relatively easy one. It is simple enough to perform an exhaustive search on it to reveal the true Pareto front, which is shown in Figure 9a. The corresponding paths are then shown in Figure 9b. The Pareto front is built from three solutions, with one (the longest one but with the least number of necessary turns), three (the shortest one but with the most number of necessary turns), and two (compromise) TPs. The first solution can be described as the easiest to control (purple curve in Figure 9b), while the third one (red curve in Figure 9b) is the fastest one. With the information about the trade-offs between objectives, the final path can be than selected.

The convergence of the VNDMOPSO is compared against the VNDGDE3, VLMOPSO, and SCMOPSO algorithms. The quality of the found solutions is expressed in terms of the dHV metric defined in (9). Nevertheless, the true Pareto front is not known for the pathfinding problems. Therefore, $\mathrm{HV}^{\mathcal{PF}}$ is computed for a combined non-dominated set created from the union of all runs of the four algorithms (every algorithm was tested 100 times). In addition, the reference point necessary for the calculation of individual HV values was taken as the combination of the worst values in individual objective values for the combined non-dominated set.

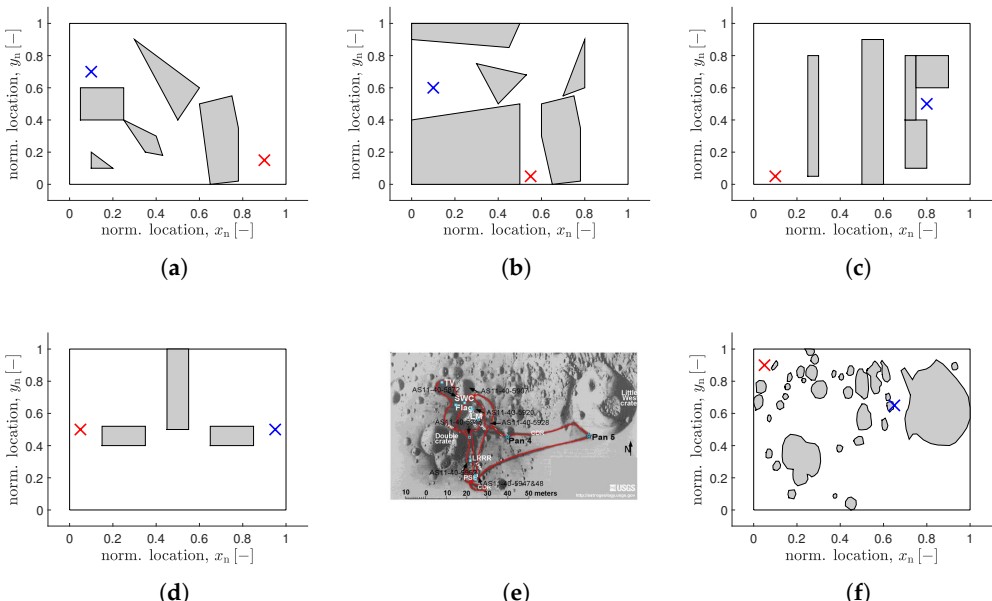

**Figure 8.** Maps of obstacles for: (**a–d**) artificial problem instances $\mathcal{P}1$–$\mathcal{P}4$, (**e**) real-life photograph of Apollo 11 landing area [46], and (**f**) instance $\mathcal{P}5$ based on (**e**). Markers '×' show locations of the robot's starting point, $r_\mathcal{S}$ (red color), and target point, $r_\mathcal{T}$ (blue).

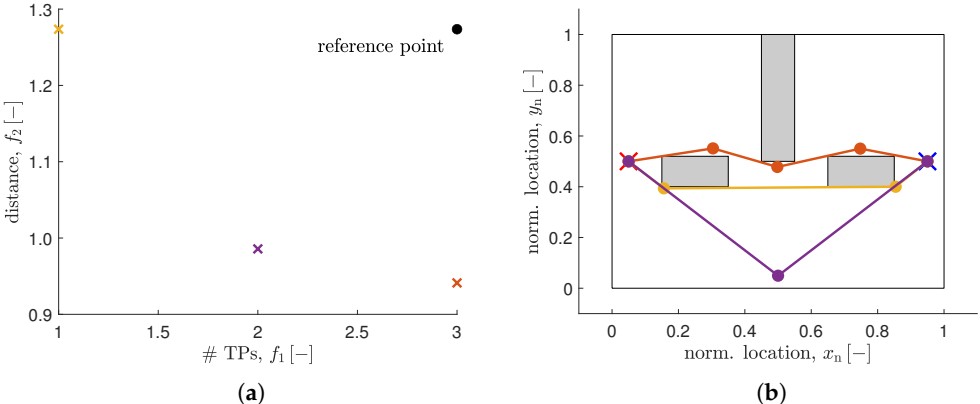

**Figure 9.** The true solution of the $\mathcal{P}4$ problem instance found by an exhaustive search: (**a**) the Pareto front formed by three discrete points (marked by symbols "×"); and (**b**) the found paths for corresponding members of the Pareto front.

The dHV results are shown in form of boxplots in Figure 10. The VNDMOPSO clearly outperforms the other three algorithms in all the pathfinding problem instances $\mathcal{P}1$–$\mathcal{P}5$. This can be explained by a proper balance of the convergence speed and the robustness provided by optimal setting of the $p_g$ and $p_b$ parameters of VNDMOPSO. VLMOPSO also outperforms the VNDGDE3 algorithm. This is probably due to the fact that the PSO algorithm, due to its general approach for updating the position of agents, is generally more suitable for solving problems that seek the optimal locations of a set of components. The SCMOPSO achieves the worst results from the whole tested set of algorithms. This is probably caused by the fact that particles in SCMOPSO can learn only from particles within the same class and not from the whole swarm. Therefore, running SCMOPSO resembles more the situation when we would run a conventional MOPSO for each possible dimension separately but with a limited number of agents.

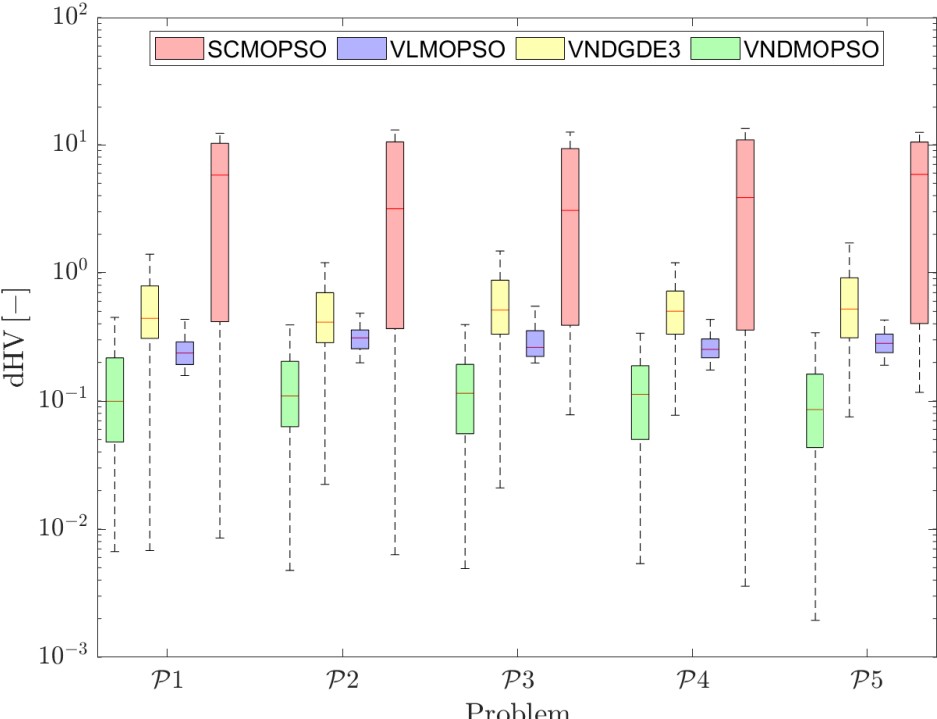

**Figure 10.** The standard boxplots (min, 1st quartile, median, 3rd quartile, max value) of the dHV metric for algorithms VNDMOPSO, VNDGDE3, VLMOPSO, and SCMOPSO on five pathfinding problem instances $\mathcal{P}1$–$\mathcal{P}5$.

Figure 11 replicates the maps of five pathfinding problem instances $\mathcal{P}1$–$\mathcal{P}5$. This time, a final path from the Pareto-optimal set found by the VNDMOPSO algorithm is highlighted with a red curve. The path corresponding to the DSV with the maximal value of the hypervolume metric, $u_{\mathrm{HV}}$, is selected from the Pareto-optimal set. Such a solution dominates the largest portion of the objective space and therefore should not prioritize any of the objectives: the minimal length of the path, and the minimal number of turning points.

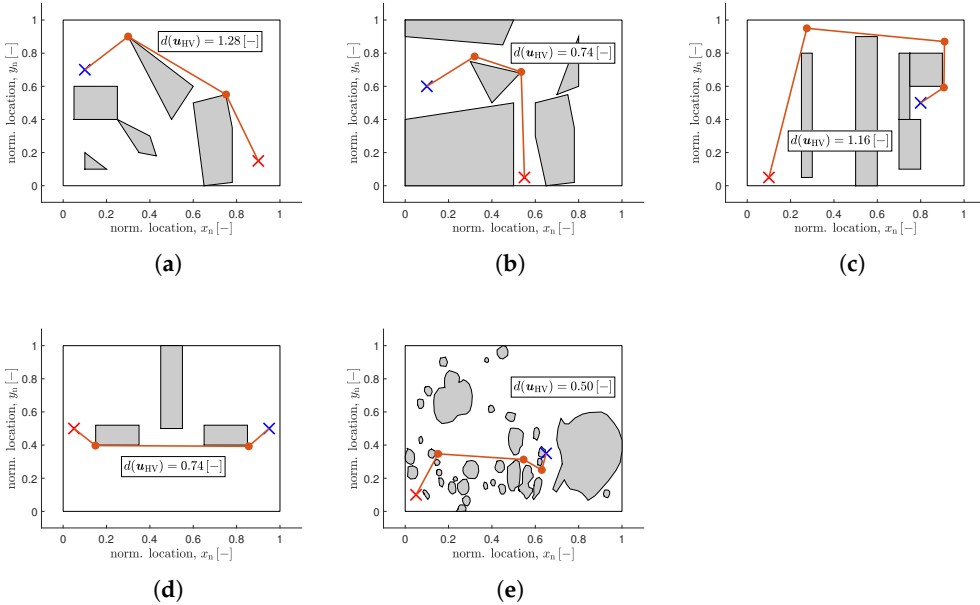

**Figure 11.** Selected Pareto-optimal solutions found by VNDMOPSO algorithm with maximal value of the HV metric for pathfinding problem instances $\mathcal{P}1$–$\mathcal{P}5$ (**a**–**e**).

## 5. Conclusions

This paper describes the full evolution of a novel multi-objective particle swarm optimization algorithm with a variable number of dimensions. According to our knowledge, VNDMOPSO is the first PSO-based algorithm that is able to solve VND MO problems and that shares the information of individual variables among candidate solutions that have different dimensions without enlarging the size of the decision space vectors by additional variables. Algorithm VNDMOPSO was compared with two other state-of-the-art algorithms, namely VLMOPSO and VNDGDE3, on a suite of 21 benchmark problems. It was shown that the convergence properties of the VNDMOPSO algorithm are superior to the VLMOPSO algorithm and at least comparable to the convergence of VNDGDE3. Then, the influence of the only new controlling parameter (the probability of dimension transition) was assessed and the recommended values are given in the paper. Finally, VNDMOPSO was used to solve five space robot pathfinding problem instances and showed significantly better results compared with VLMOPSO, SCMOPSO, and VNDGDE3. Moreover, the SRPP has been formulated as a VND MO problem, which, on the one hand, brings better information about necessary compromises to the decision about the final path of the robot, but, on the other hand, it does not force the user to limit the route optimization process in any way—either in terms of the number of turning points or a priori setting of weights to individual objectives. Moreover, the proposed method allows adding any additional objective, e.g., to minimize the probability of collision and others, easily.

**Supplementary Materials:** The following supporting information can be downloaded at: https://www.mdpi.com/article/10.3390/a16060307/s1. Supplementary material has been entitled "Definition of Multi-objective Benchmark Problems with Variable Number of Dimensions" labeled as S1. It contains the definition of all the benchmark problems used in this study to assess the convergence properties of the VNDMOPSO algorithm. This material cites references [32,38,42–45].

**Funding:** The research described in this paper was supported by the Internal Grant Agency of the Brno University of Technology under project no. FEKT-S-23-8191.

**Data Availability Statement:** The data that support the findings of this study are available upon request at: kadlecp@vut.cz.

**Conflicts of Interest:** The author declares no conflict of interest.

## Abbreviations

The following abbreviations are used in this manuscript:

| | |
|---|---|
| DE | Differential evolution |
| dHV | Distance hypervolume metric |
| DSV | Decision space vector |
| EOA | Evolutionary optimization algorithm |
| GDE3 | Generalized differential evolution |
| HA | Heuristic algorithm |
| HV | Hypervolume metric |
| MO | Multi-objective |
| MOOP | Multi-objective optimization problem |
| MOPSO | Multi-objective particle swarm optimization |
| PSO | Particle swarm optimization |
| SCMOPSO | Social class MOPSO |
| SRPP | Space robot pathfinding problem |
| TP | Turning (way) point |
| VLMOPSO | Variable length MO PSO |
| VND | Variable number of dimensions |
| VNDGDE3 | Variable number of dimensions GDE3 |
| VNDMOPSO | Variable number of dimensions MOPSO |

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
