# Peer review of "Multi-Objective PSO with Variable Number of Dimensions for Space Robot Path Optimization"

_algorithms, doi:10.3390/a16060307_

Round 1

Reviewer 1 Report

In the manuscript entitled "Multi-objective PSO with Variable Number of Dimensions for Space Robot Path Optimization" concern the solution pathfinding problem. The manuscript is included in the aim and scope of the Algorithms journal. This is because describe the adaptation of the particle swarm optimization (PSO) algorithm to solve path of motion of robot. The algorithm has been carefully prepared by the author, only some editing modifications are necessary. The manuscript is well structured and methodology is described by the clear way.

The detailed comments and remarks to the Author are described below:

1.      The first comment is to the title of the manuscript "Multi-objective", the author should underline the multi-objective of path planning. According to me the multi-objective problem of path planning can take into account: (a) the length of the path, (b) the total time of travel, (c) fuel/energy cost. Please underline why your problem is multi-objective.

2.      The word "Evolutionary Optimization" should be replaced by "heuristic algorithms". The evolutionary algorithms suit to evolutionary algorithms (genetic algorithm, evolutionary algorithms). But PSO belong to heuristic algorithms (please read: Performance analysis of selected metaheuristic optimization algorithms applied in the solution of an unconstrained task).

3.      The row 30, the "single objective" should be written single-objective.

4.      In the rows 25-28 the authors wrote "Power consumption and route reliability play at least the same role as the length of the route itself due to the unavailability of replacement sources and the very limited (and costly) or complete impossibility of repairing the device in the event of a collision with an obstacle". But according to me, the all this parameters should be taken into account in multi-objective function.

5.      In the row 78, the author wrote "Social class PSO algorithm was introduced". In general PSO algorithm was elaborated on the social interaction in bird flocking and fish shoal. This statement can be not clear, it should be deepened in the introduction.

6.      At the end of the introduction paragraph, the author can add a short section describing information about content of all chapters.

7.      In the row 118, the subscript "d" should be written by italic font.

8.      In equation (6) the author multiples the time steep by the vector of particle position the unit of multiplication is m*m/s=m^2/s. Please correct this equation.

9.      In the many places of manuscript, the subscript in (Ng, Nb, Nu) should be written by italic font.

10.  In the row 371 author wrote "All the algorithms are implemented using a MATLAB–based toolbox called Fast Optimization Procedures(FOPS)". The important question is, the all used algorithms are part of Matlb software, or author developed al optimization software using Maltab environment?

11.  The quality of Figures in Figure 3 must be improved before final publication.

12.  The Figure 7(b) is described by the authors as a possible Pareto front. We see three different path, but in the theory of optimization we can describe as: fast, easy and motion control complicated. Please discuss your explanation.

13.  The conclusion should underline the novelty.

The minor editing language is required. The correct "single optimization" is single-optimization.

Author Response

Thank you very much for your recommendations! The point-by-point responses with the description of the corresponding manuscript changes can be found in the attached *.pdf file.

Reviewer 2 Report

The author describes a spatial pathfinding problem in the form of a multi-objective optimization problem. Overall, the text is clear and easy to follow (there are a few typos but nothing significant other than that).

- Figure 3 is kind of hard to read and infer conclusions from it. 

- Although it is understood that Table 4 is needed to showcase how each algorithm compares to the rest, it is a lot of information packed in multiple rows. Maybe replace it with a more concise table that briefly summarizes the + and - cases.

- (Very minor) Figure 6 could be shown a second time with the path that the proposed framework generated illustrated on the initial problems.

- A more serious concern is that the author compares their work with a previous work of theirs. To make this a more sound publication, it would be best to include state-of-art frameworks from different labs and collaborators.

Author Response

(The authors gave the same response as above.)

Round 2

Reviewer 2 Report

I am satisfied with the revision.